# Unsupervised Learning of Artistic Styles with Archetypal Style Analysis

**Daan Wynen, Cordelia Schmid, Julien Mairal**
Univ. Grenoble Alpes, Inria, CNRS, Grenoble INP,* LJK, 38000 Grenoble, France
`firstname.lastname@inria.fr`

## Abstract

In this paper, we introduce an unsupervised learning approach to automatically discover, summarize, and manipulate artistic styles from large collections of paintings. Our method is based on archetypal analysis, which is an unsupervised learning technique akin to sparse coding with a geometric interpretation. When applied to neural style representations from a collection of artworks, it learns a dictionary of archetypal styles, which can be easily visualized. After training the model, the style of a new image, which is characterized by local statistics of deep visual features, is approximated by a sparse convex combination of archetypes. This enables us to interpret which archetypal styles are present in the input image, and in which proportion. Finally, our approach allows us to manipulate the coefficients of the latent archetypal decomposition, and achieve various special effects such as style enhancement, transfer, and interpolation between multiple archetypes.

## 1 Introduction

Artistic style transfer consists in manipulating the appearance of an input image such that its semantic content and its scene organization are preserved, but a human may perceive the modified image as having been painted in a similar fashion as a given target painting. Closely related to previous approaches to texture synthesis based on modeling statistics of wavelet coefficients [8, 19], the seminal work of Gatys et al. [5, 6] has recently shown that a deep convolutional neural network originally trained for classification tasks yields a powerful representation for style and texture modeling. Specifically, the description of "style" in [5] consists of local statistics obtained from deep visual features, represented by the covariance matrices of feature responses computed at each network layer. Then, by using an iterative optimization procedure, the method of Gatys et al. [5] outputs an image whose deep representation should be as close as possible to that of the input content image, while matching the statistics of the target painting. This approach, even though relatively simple, leads to impressive stylization effects that are now widely deployed in consumer applications.

Subsequently, style transfer was improved in many aspects. First, removing the relatively slow optimization procedure of [5] was shown to be possible by instead training a convolutional neural network to perform style transfer [10, 22]. Once the model has been learned, stylization of a new image requires a single forward pass of the network, allowing real-time applications. Whereas these networks were originally trained to transfer a single style (*e.g.*, a network trained for producing a "Van Gogh effect" was unable to produce an image resembling Monet's paintings), recent approaches have been able to train a convolutional neural network to transfer multiple styles from a collection of paintings and to interpolate between styles [1, 4, 9].

Then, key to our work, Li et al. [11] recently proposed a simple learning-free and optimization-free procedure to modify deep features of an input image such that their local statistics approximately

match those of a target style image. Their approach is based on decoders that have been trained to invert a VGG network [21], allowing them to iteratively whiten and recolor feature maps of every layer, before eventually reconstructing a stylized image for any arbitrary style. Even though the approach may not perserve content details as accurately as other learning-based techniques [24], it nevertheless produces ousanding results given its simplicity and universality. Finally, another trend consists of extending style transfer to other modalities such as videos [20] or natural photographs [12] (to transfer style from photo to photo instead of painting to photograph).

Whereas the goal of these previous approaches was to improve style transfer, we address a different objective and propose to use an unsupervised method to learn style representations from a potentially large collection of paintings. Our objective is to automatically discover, summarize, and manipulate artistic styles present in the collection. To achieve this goal, we rely on a classical unsupervised learning technique called archetypal analysis [3]; despite its moderate popularity, this approach is related to widely-used paradigms such as sparse coding [13, 15] or non-negative matrix factorization [16]. The main advantage of archetypal analysis over these other methods is mostly its better interpretability, which is crucial to conduct applied machine learning work that requires model interpretation.

In this paper, we learn archetypal representations of style from image collections. Archetypes are simple to interpret since they are related to convex combinations of a few image style representations from the original dataset, which can thus be visualized (see, *e.g.*, [2] for an application of archetypal analysis to image collections). When applied to painter-specific datasets, they may for instance capture the variety and evolution of styles adopted by a painter during his career. Moreover, archetypal analysis offers a dual interpretation view: if on the one hand, archetypes can be seen as convex combinations of image style representations from the dataset, each image's style can also be decomposed into a convex combination of archetypes on the other hand. Then, given an image, we may automatically interpret which archetypal style is present in the image and in which proportion, which is a much richer information than what a simple clustering approach would produce. When applied to rich data collections, we sometimes observe trivial associations (*e.g.*, the image's style is very close to one archetype), but we also discover meaningful interesting ones (when an image's style may be interpreted as an interpolation between several archetypes).

After establishing archetypal analysis as a natural tool for unsupervised learning of artistic style, we also show that it provides a latent parametrization allowing to manipulate style by extending the universal style transfer technique of [11]. By changing the coefficients of the archetypal decomposition (typically of small dimension, such as 256) and applying stylization, various effects on the input image may be obtained in a flexible manner. Transfer to an archetypal style is achieved by selecting a single archetype in the decomposition; style enhancement consist of increasing the contribution of an existing archetype, making the input image more "archetypal". More generally, exploring the latent space allows to create and use styles that were not necessarily seen in the dataset, see Figure 1.

To the best of our knowledge, [7] is the closest work to ours in terms of latent space description of style; our approach is however based on significantly different tools and our objective is different. Whereas a latent space is learned in [7] for style description in order to improve the generalization of a style transfer network to new unseen paintings, our goal is to build a latent space that is directly interpretable, with one dimension associated to one archetypal style.

The paper is organized as follows: Section 2 presents the archetypal style analysis model, and its application to a large collection of paintings. Section 3 shows how we use them for various style manipulations. Finally, Section 4 is devoted to additional experiments and implementation details.

## 2   Archetypal Style Analysis

In this section, we show how to use archetypal analysis to learn style from large collections of paintings, and subsequently perform style decomposition on arbitrary images.

**Learning a latent low-dimensional representation of style.**   Given an input image, denoted by $I$, we consider a set of feature maps $\mathbf{F}_1, \mathbf{F}_2, \ldots, \mathbf{F}_L$ produced by a deep network. Following [11], we consider five layers of the VGG-19 network [21], which has been pre-trained for classification. Each feature map $\mathbf{F}_l$ may be seen as a matrix in $\mathbb{R}^{p_l \times m_l}$ where $p_l$ is the number of channels and $m_l$ is the number of pixel positions in the feature map at layer $l$. Then, we define the style of $I$ as the collection

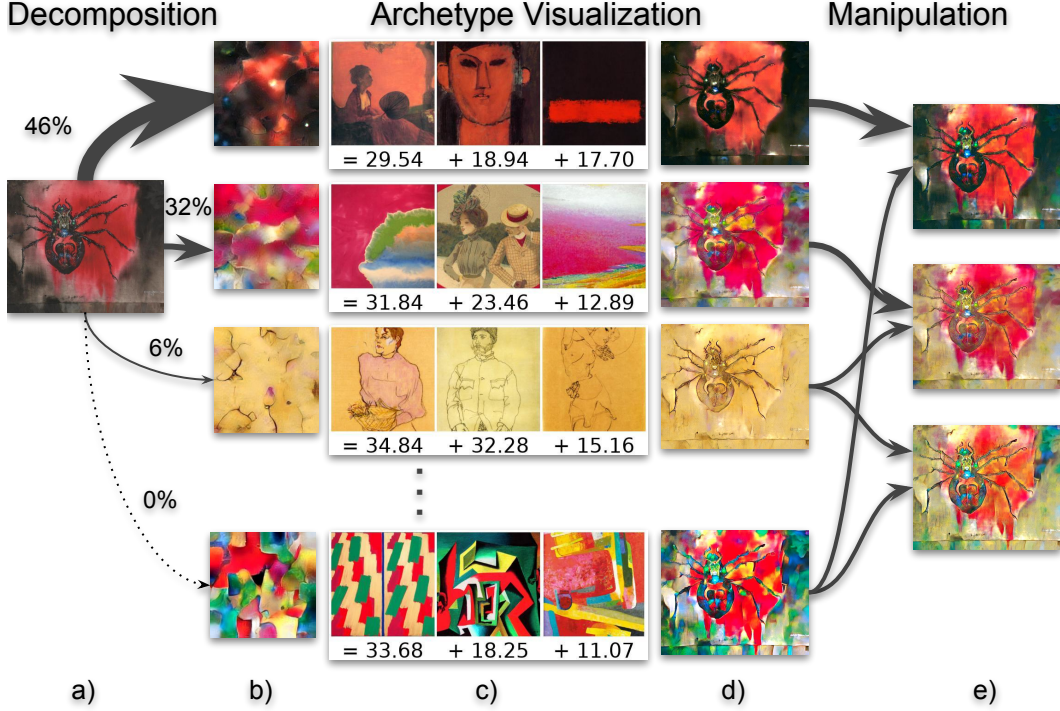

**Decomposition**  **Archetype Visualization**  **Manipulation**

46%

32%

6%

0%

= 29.54  + 18.94  + 17.70

= 31.84  + 23.46  + 12.89

= 34.84  + 32.28  + 15.16

= 33.68  + 18.25  + 11.07

a)      b)      c)      d)      e)

Figure 1: Using deep archetypal style analysis, we can represent an artistic image (a) as a convex combination of archetypes. The archetypes can be visualized as synthesized textures (b), as a convex combination of artworks (c) or, when analyzing a specific image, as stylized versions of that image itself (d). Free recombination of the archetypal styles then allows for novel stylizations of the input.

of first-order and second order statistics $\{\boldsymbol{\mu}_1, \boldsymbol{\Sigma}_1, \ldots, \boldsymbol{\mu}_L, \boldsymbol{\Sigma}_L\}$ of the feature maps, defined as

$$\boldsymbol{\mu}_l = \tfrac{1}{m_l} \sum_{j=1}^{m_l} \mathbf{F}_l[j] \in \mathbb{R}^{p_l} \quad \text{and} \quad \boldsymbol{\Sigma}_l = \tfrac{1}{m_l} \sum_{j=1}^{m_l} (\mathbf{F}_l[j] - \boldsymbol{\mu}_l)(\mathbf{F}_l[j] - \boldsymbol{\mu}_l)^\top \in \mathbb{R}^{p_l \times p_l},$$

where $\mathbf{F}_l[j]$ represents the column in $\mathbb{R}^{p_l}$ that carries the activations at position $j$ in the feature map $\mathbf{F}_l$. A style descriptor is then defined as the concatenation of all parameters from the collection $\{\boldsymbol{\mu}_1, \boldsymbol{\Sigma}_1, \ldots, \boldsymbol{\mu}_L, \boldsymbol{\Sigma}_L\}$, normalized by the number of parameters at each layer—that is, $\boldsymbol{\mu}_l$ and $\boldsymbol{\Sigma}_l$ are divided by $p_l(p_l + 1)$, which was found to be empirically useful for preventing layers with more parameters to be over-represented. The resulting vector is very high-dimensional, but it contains key information for artistic style [5]. Then, we apply a singular value decomposition on the style representations from the paintings collection to reduce the dimension to $4\,096$ while keeping more than 99% of the variance. Next, we show how to obtain a lower-dimensional latent representation.

**Archetypal style representation.** Given a set of vectors $\mathbf{X} = [\mathbf{x}_1, \ldots, \mathbf{x}_n]$ in $\mathbb{R}^{p \times n}$, archetypal analysis [3] learns a set of archetypes $\mathbf{Z} = [\mathbf{z}_1, \ldots, \mathbf{z}_k]$ in $\mathbb{R}^{p \times k}$ such that each sample $\mathbf{x}_i$ can be well approximated by a convex combination of archetypes—that is, there exists a code $\boldsymbol{\alpha}_i$ in $\mathbb{R}^k$ such that $\mathbf{x}_i \approx \mathbf{Z}\boldsymbol{\alpha}_i$, where $\boldsymbol{\alpha}_i$ lies in the simplex

$$\Delta_k = \Big\{\boldsymbol{\alpha} \in \mathbb{R}^k \ \text{s.t.} \ \boldsymbol{\alpha} \geq 0 \ \text{and} \ \textstyle\sum_{j=1}^k \boldsymbol{\alpha}[j] = 1\Big\}.$$

Conversely, each archetype $\mathbf{z}_j$ is constrained to be in the convex hull of the data and there exists a code $\boldsymbol{\beta}_j$ in $\Delta_n$ such that $\mathbf{z}_j = \mathbf{X}\boldsymbol{\beta}_j$. The natural formulation resulting from these geometric constraints is then the following optimization problem

$$\min_{\substack{\boldsymbol{\alpha}_1, \ldots, \boldsymbol{\alpha}_n \in \Delta_k \\ \boldsymbol{\beta}_1, \ldots, \boldsymbol{\beta}_k \in \Delta_n}} \frac{1}{n} \sum_{i=1}^n \|\mathbf{x}_i - \mathbf{Z}\boldsymbol{\alpha}_i\|^2 \quad \text{s.t.} \quad \mathbf{z}_j = \mathbf{X}\boldsymbol{\beta}_j \quad \text{for all } j = 1, \ldots, k, \tag{1}$$

which can be addressed efficiently with dedicated solvers [2]. Note that the simplex constraints lead to non-negative sparse codes $\boldsymbol{\alpha}_i$ for every sample $\mathbf{x}_i$ since the simplex constraint enforces the vector $\boldsymbol{\alpha}_i$

to have unit $\ell_1$-norm, which has a sparsity-inducing effect [13]. As a result, a sample $\mathbf{x}_i$ will be associated in practice to a few archetypes, as observed in our experimental section. Conversely, an archetype $\mathbf{z}_j = \mathbf{X}\boldsymbol{\beta}_j$ can be represented by a non-negative sparse code $\boldsymbol{\beta}_j$ and thus be associated to a few samples corresponding to non-zero entries in $\boldsymbol{\beta}_j$.

In this paper, we use archetypal analysis on the $4\,096$-dimensional style vectors previously described, and typically learn between $k = 32$ to $k = 256$ archetypes. Each painting's style can then be represented by a sparse low-dimensional code $\boldsymbol{\alpha}$ in $\Delta_k$, and each archetype is itself associated to a few input paintings, which is crucial for their interpretation (see the experimental section). Given a fixed set of archetypes $\mathbf{Z}$, we may also quantify the presence of archetypal styles in a new image $I$ by solving the convex optimization problem

$$\boldsymbol{\alpha}^\star \in \underset{\boldsymbol{\alpha} \in \Delta_k}{\arg\min} \, \|\mathbf{x} - \mathbf{Z}\boldsymbol{\alpha}\|^2, \tag{2}$$

where $\mathbf{x}$ is the high-dimensional input style representation described at the beginning of this section. Encoding an image style into a sparse vector $\boldsymbol{\alpha}$ allows us to obtain interesting interpretations in terms of the presence and quantification of archetypal styles in the input image. Next, we show how to manipulate the archetypal decomposition by modifying the universal feature transform of [11].

## 3   Archetypal Style Manipulation

In the following, we briefly present the universal style transfer approach of [11] and introduce a modification that allows us to better preserve the content details of the original images, before presenting how to use the framework for archetypal style manipulation.

**A new variant of universal style transfer.**   We assume, in this section only, that we are given a content image $I^c$ and a style image $I^s$. We also assume that we are given pairs of encoders/decoders $(d_l, e_l)$ such that $e_l(I)$ produces the $l$-th feature map previously selected from the VGG network and $d_l$ is a decoder that has been trained to approximately "invert" $e_l$—that is, $d_l(e_l(I)) \approx I$.

Universal style transfer builds upon a simple idea. Given a "content" feature map $\mathbf{F}_c$ in $\mathbb{R}^{p \times m}$, making local features match the mean and covariance structure of another "style" feature map $\mathbf{F}_s$ can be achieved with simple whitening and coloring operations, leading overall to an affine transformation:

$$C^s(\mathbf{F}_c) := \mathbf{C}^s \mathbf{W}^c (\mathbf{F}^c - \boldsymbol{\mu}^c) + \boldsymbol{\mu}^s,$$

where $\boldsymbol{\mu}^c, \boldsymbol{\mu}^s$ are the mean of the content and style feature maps, respectively, $\mathbf{C}^s$ is the coloring matrix and $\mathbf{W}^c$ is a whitening matrix that decorrelates the features. We simply summarize this operation as a single function $C^s : \mathbb{R}^{p \times m} \to \mathbb{R}^{p \times m}$.

Of course, feature maps between network layers are interconnected and such coloring and whitening operations cannot be applied simultaneously at every layer. For this reason, the method produces a sequence of stylized images $\hat{I}_l$, one per layer, starting from the last one $l = L$ in a coarse-to-fine manner, and the final output is $\hat{I}_1$. Given a stylized image $\hat{I}_{l+1}$ (with $\hat{I}_{L+1} = I^c$), we propose the following update, which differs slightly from [11], for a reason we will detail below:

$$\hat{I}_l = d_l \left( \gamma \left( \delta C_l^s(e_l(\hat{I}_{l+1})) + (1-\delta) C_l^s(e_l(I^c)) \right) + (1-\gamma) e_l(I^c) \right), \tag{3}$$

where $\gamma$ in $(0, 1)$ controls the amount of stylization since $e_l(I^c)$ corresponds to the $l$-th feature map of the original content image. The parameter $\delta$ in $(0, 1)$ controls how much one should trust the current stylized image $\hat{I}_{l+1}$ in terms of content information before stylization at layer $l$. Intuitively,
(a) $d_l(C_l^s(e_l(\hat{I}_{l+1})))$ can be interpreted as a refinement of the stylized image at layer $l+1$ in order to take into account the mean and covariance structure of the image style at layer $l$.
(b) $d_l(C_l^s(e_l(I^c)))$ can be seen as a stylization of the content image by looking at the correlation/mean structure of the style at layer $l$ regardless of the structure at the preceding stylization steps.

Whereas $\hat{I}_{l+1}$ takes into account the style structure of the top layers, it may also have lost a significant amount of content information, in part due to the fact that the decoders $d_l$ do not invert perfectly the encoders and do not correctly recover fine details. For this reason, being able to make a trade-off between (a) and (b) to explicitly use the original content image $I^c$ at each layer is important.

In contrast, the update of [11] involves a single parameter $\gamma$ and is of the form

$$\hat{I}_l = d_l \left( \gamma \left( C_l^s(e_l(\hat{I}_{l+1})) \right) + (1 - \gamma)e_l(\hat{I}_{l+1}) \right). \tag{4}$$

Notice that here the original image $I^c$ is used only once at the beginning of the process, and details that have been lost at layer $l + 1$ have no chance to be recovered at layer $l$. We present in the experimental section the effect of our variant. Whenever one is not looking for a fully stylized image—that is, $\gamma < 1$ in (3) and (4), content details can be much better preserved with our approach.

**Archetypal style manipulation.** We now aim to analyze styles and change them in a controllable manner based on styles present in a large collection of images rather than on a single image. To this end, we use the archetypal style analysis procedure described in Section 2. Given now an image $I$, its style, originally represented by a collection of statistics $\{\boldsymbol{\mu}_1, \boldsymbol{\Sigma}_1, \dots, \boldsymbol{\mu}_L, \boldsymbol{\Sigma}_L\}$, is approximated by a convex combination of archetypes $[\mathbf{z}_1, \dots, \mathbf{z}_k]$, where archetype $\mathbf{z}_j$ can also be seen as the concatenation of statistics $\{\boldsymbol{\mu}_1^j, \boldsymbol{\Sigma}_1^j, \dots, \boldsymbol{\mu}_L^j, \boldsymbol{\Sigma}_L^j\}$. Indeed, $\mathbf{z}_j$ is associated to a sparse code $\boldsymbol{\beta}_j$ in $\Delta_n$, where $n$ is the number of training images—allowing us to define for archetype $j$ and layer $l$

$$\boldsymbol{\mu}_l^j = \sum_{i=1}^n \boldsymbol{\beta}_j[i]\boldsymbol{\mu}_l^{(i)} \quad \text{and} \quad \boldsymbol{\Sigma}_l^j = \sum_{i=1}^n \boldsymbol{\beta}_j[i]\boldsymbol{\Sigma}_l^{(i)},$$

where $\boldsymbol{\mu}_l^{(i)}$ and $\boldsymbol{\Sigma}_l^{(i)}$ are the mean and covariance matrices of training image $i$ at layer $l$. As a convex combination of covariance matrices, $\boldsymbol{\Sigma}_l^j$ is positive semi-definite and can be also interpreted as a valid covariance matrix, which may then be used for a coloring operation producing an "archetypal" style.

Given now a sparse code $\boldsymbol{\alpha}$ in $\Delta_k$, a new "style" $\{\hat{\boldsymbol{\mu}}_1, \hat{\boldsymbol{\Sigma}}_1, \dots, \hat{\boldsymbol{\mu}}_L, \hat{\boldsymbol{\Sigma}}_L\}$ can be obtained by considering the convex combination of archetypes:

$$\hat{\boldsymbol{\mu}}_l = \sum_{j=1}^k \boldsymbol{\alpha}[j]\boldsymbol{\mu}_l^j \quad \text{and} \quad \hat{\boldsymbol{\Sigma}}_l = \sum_{j=1}^k \boldsymbol{\alpha}[j]\boldsymbol{\Sigma}_l^j.$$

Then, the collection of means and covariances $\{\hat{\boldsymbol{\mu}}_1, \hat{\boldsymbol{\Sigma}}_1, \dots, \hat{\boldsymbol{\mu}}_L, \hat{\boldsymbol{\Sigma}}_L\}$ may be used to define a coloring operation. Three practical cases come to mind: (i) $\boldsymbol{\alpha}$ may be a canonical vector that selects a single archetype; (ii) $\boldsymbol{\alpha}$ may be any convex combination of archetypes for archetypal style interpolation; (iii) $\boldsymbol{\alpha}$ may be a modification of an existing archetypal decomposition to enhance a style already present in an input image $I$—that is, $\boldsymbol{\alpha}$ is a variation of $\boldsymbol{\alpha}^\star$ defined in (2).

## 4 Experiments

In this section, we present our experimental results on two datasets described below. Our implementation is in PyTorch [17] and relies in part on the universal style transfer implementation[2]. Archetypal analysis is performed using the SPAMS software package [2, 14], and the singular value decomposition is performed by scikit-learn [18]. Our implementation will be made publicly available. Further examples can be found at `http://pascal.inrialpes.fr/data2/archetypal_style`.

**GanGogh** is a collection of 95997 artworks[3] downloaded from WikiArt.[4] The images cover most of the freely available WikiArt catalog, with the exception of artworks that are not paintings. Due to the collaborative nature of WikiArt, there is no guarantee for an unbiased selection of artworks, and the presence of various styles varies significantly. We compute 256 archetypes on this collection.

**Vincent van Gogh** As a counter point to the GanGogh collection, which spans many styles over a long period of time and has a significant bias towards certain art styles, we analyze the collection of Vincent van Gogh's artwork, also from the WikiArt catalog. Based on the WikiArt metadata, we exclude a number of works not amenable to artistic style transfer such as sketches and studies. The collection counts 1154 paintings and drawings in total, with the dates of their creation ranging from 1858 to 1926. Given the limited size of the collection, we only compute 32 archetypes.

## 4.1 Archetypal Visualization

To visualize the archetypes, we first synthesize one texture per archetype by using its style representation to repeatedly stylize an image filled with random noise, as described in [11]. We then display paintings with significant contributions. In Figure 2, we present visualizations for a few archetypes. The strongest contributions usually exhibit a common characteristic like stroke style or choice of colors. Smaller contributions are often more difficult to interpret (see supplementary material for the full set of archetypes). Figure 2a also highlights correlation between content and style: the archetype on the third row is only composed of portraits.

To see how the archetypes relate to each other, we also compute t-SNE embeddings [23] and display them with two spatial dimensions. In Figure 3, we show the embeddings for the GanGogh collection, by using the texture representation for each archetype. The middle of the figure is populated by Baroque and Renaissance styles, whereas the right side exhibits abstract and cubist styles.

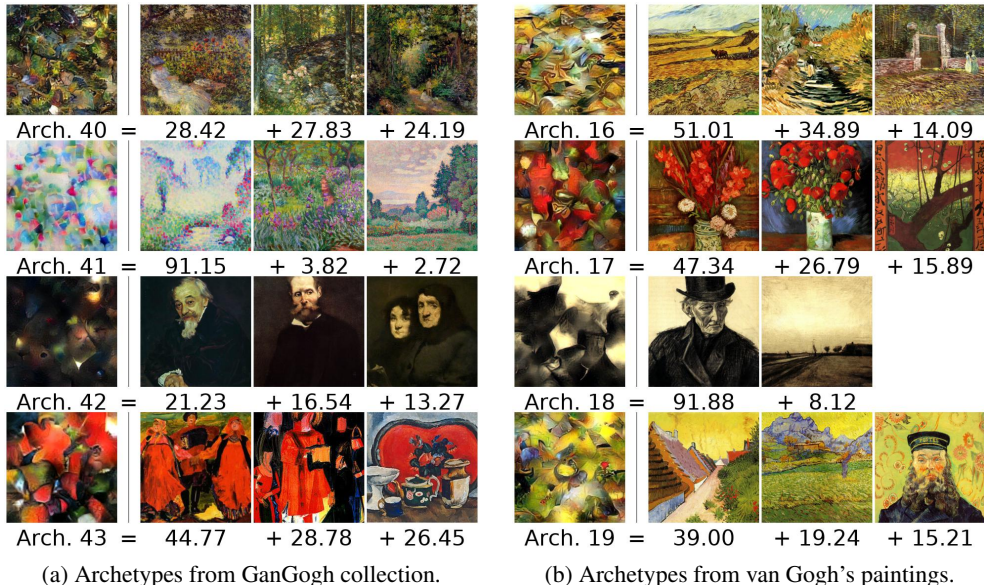

| (a) Archetypes from GanGogh collection. | (b) Archetypes from van Gogh's paintings. |

Figure 2: Archetypes learned from the GanGogh collection and van Gogh's paintings. Each row represents one archetype. The leftmost column shows the texture representations, the following columns the strongest contributions from individual images in order of descending contribution. Each image is labelled with its contribution to the archetype. For layout considerations, only the center crop of each image is shown. Best seen by zooming on a computer screen.

Similar to showing the decomposition of an archetype into its contributing images, we display in Figure 4 examples of decompositions of image styles into their contributing archetypes. Typically, only a few archetypes contribute strongly to the decomposition. Even though often interpretable, the decomposition is sometimes trivial, whenever the image's style is well described by a single archetype. Some paintings' styles also turn out to be hard to interpret, leading to non-sparse decompositions. Examples of such trivial and "failure" cases are provided in the supplementary material.

## 4.2 Archetypal Style Manipulation

First, we study the influence of the parameters $\gamma, \delta$ and make a comparison with the baseline method of [11]. Even though this is not the main contribution of our paper, this apparently minor modification yields significant improvements in terms of preservation of content details in stylized images. Besides, the heuristic $\gamma = \delta$ appears to be visually reasonable in most cases, reducing the number of effective parameters to a single one that controls the amount of stylization. The comparison between our update (3) and (4) from [11] is illustrated in Figure 5, where the goal is to transfer an archetypal style to a Renaissance painting. More comparisons on other images and illustrations with pairs of parameters $\gamma \neq \delta$, as well as a comparison of the processing workflows, are provided in the supplementary material, confirming our conclusions.

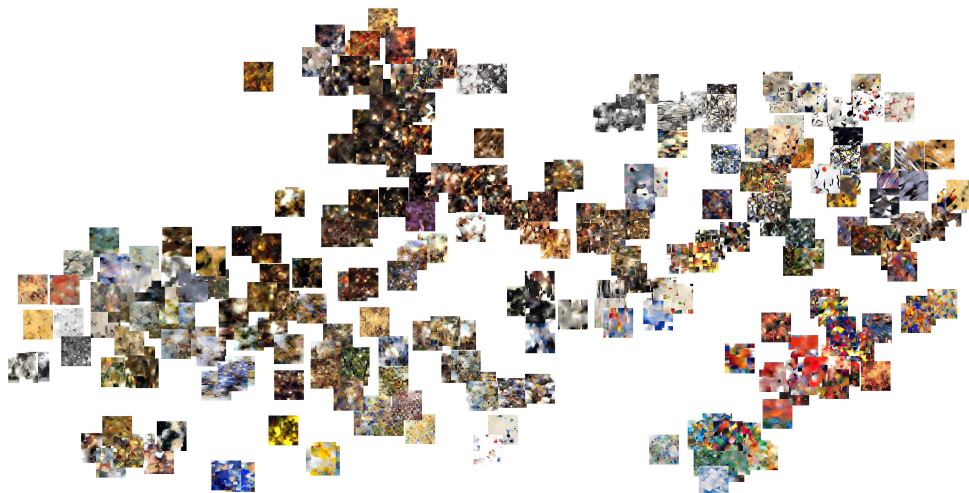

Figure 3: t-SNE embeddings of 256 archetypes computed on the GanGogh collection. Each archetype is represented by a synthesized texture. Best seen by zooming on a computer screen.

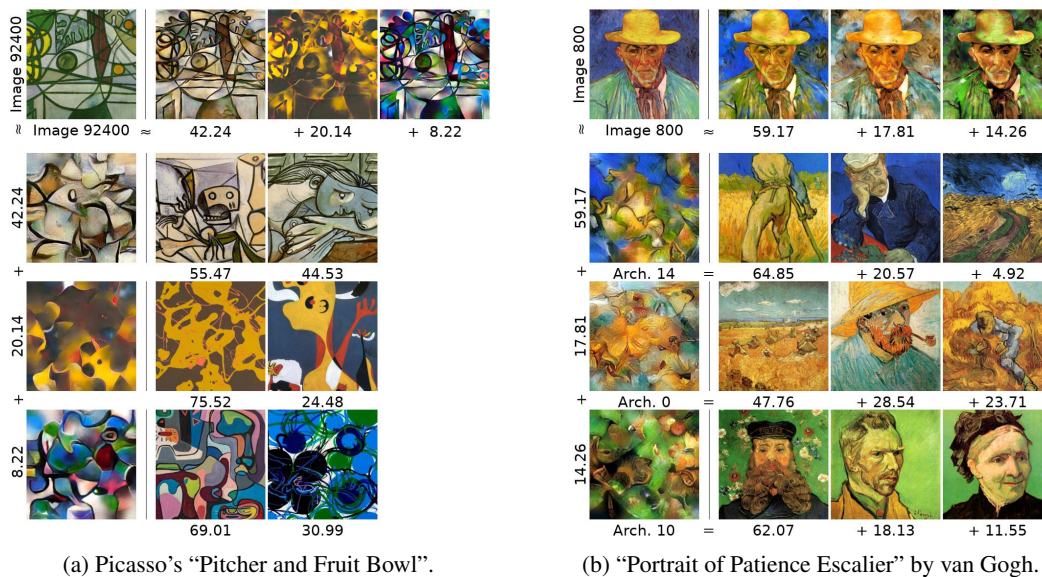

(a) Picasso's "Pitcher and Fruit Bowl".

(b) "Portrait of Patience Escalier" by van Gogh.

Figure 4: Image decompositions from the GanGogh collection and van Gogh's work. Each archetype is represented as a stylized image (top), as a texture (side) and as a decomposition into paintings.

Then, we conduct style enhancement experiments. To obtain variations of an input image, the decomposition $\alpha^\star$ of its style can serve as a starting point for stylization. Figure 6 shows the results of enhancing archetypes an image already exhibits. Intuitively, this can be seen as taking one aspect of the image, and making it stronger with respect to the other ones. In Figure 6, while increasing the contributions of the individual archetypes, we also vary $\gamma = \delta$, so that the middle image is very close visually to the original image ($\gamma = \delta = 0.5$), while the outer panels put a strong emphasis on the modified styles. As can be seen especially in the panels surrounding the middle, modifying the decomposition coefficients allows very gentle movements through the styles.

As can be seen in the leftmost and rightmost panels of Figure 6, enhancing the contribution of an archetype can produce significant changes. As a matter of fact, it is also possible, and sometimes desirable, depending on the user's objective, to manually choose a set of archetypes that are originally unrelated to the input image, and then interpolate with convex combinations of these archetypes. The results are images akin to those found in classical artistic style transfer papers. In Figure 7, we apply for instance combinations of freely chosen archetypes to "The Bitter Drunk". Other examples involving stylizing natural photographs are also provided in the supplementary material.

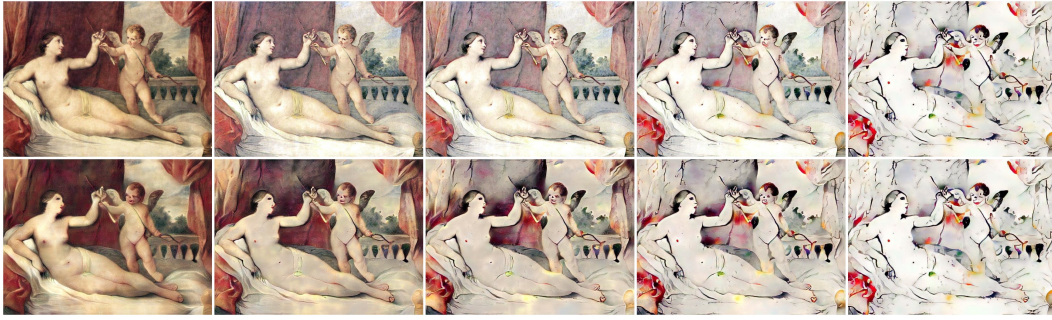

Figure 5: Top: stylization with our approach for $\gamma = \delta$, varying the product $\gamma\delta$ from $0$ to $1$ on an equally-spaced grid. Bottom: results using [11], varying $\gamma$. At $\gamma = \delta = 1$, the approaches are equivalent, resulting in equal outputs. Otherwise however, especially for $\gamma = \delta = 0$, [11] produces strong artifacts. These are *not* artifacts of stylization, since in this case, no actual stylization occurs. Rather, they are the effect of repeated, lossy encoding and decoding, since no decoder can recover information lost in a previous one. Best seen on a computer screen.

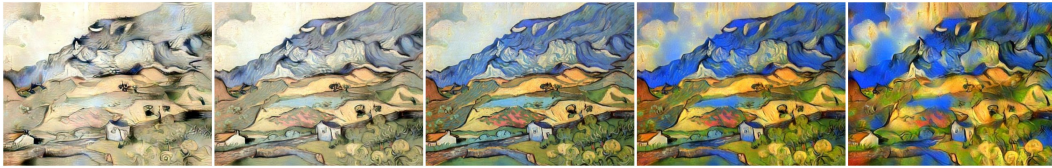

(a) "Les Alpilles, Mountain Landscape near South-Reme" by van Gogh, from the van Gogh collection.

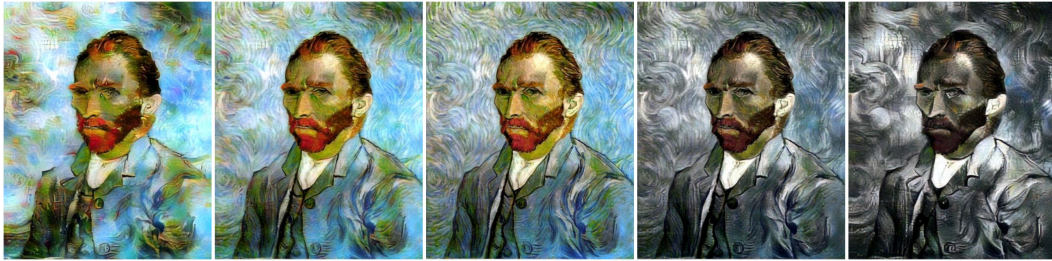

(b) Self-Portrait by van Gogh, from the van Gogh collection.

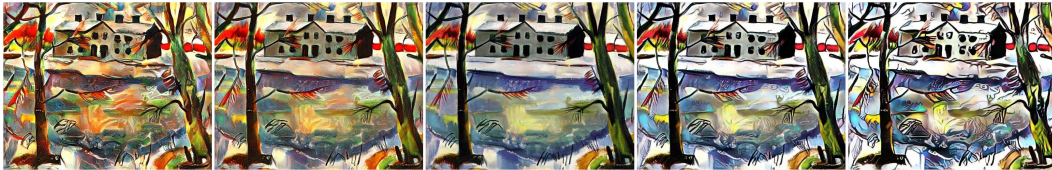

(c) "Schneeschmelze" by Max Pechstein, from the GanGogh collection.

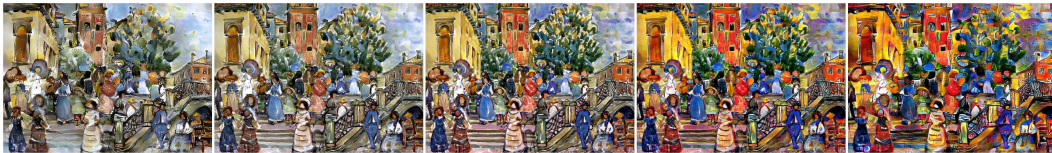

(d) "Venice" by Maurice Prendergast, from the GanGogh collection.

Figure 6: We demonstrate the enhancement of the two most prominent archetypal styles for different artworks. The middle panel shows a near-perfect reconstruction of the original content image in every case and uses parameters $\gamma, \delta = 0.5$. Then, we increase the relative weight of the strongest component towards the left, and of the second component towards the right. Simultaneously, we increase $\gamma$ and $\delta$ from $0.5$ in the middle panel to $0.95$ on the outside.

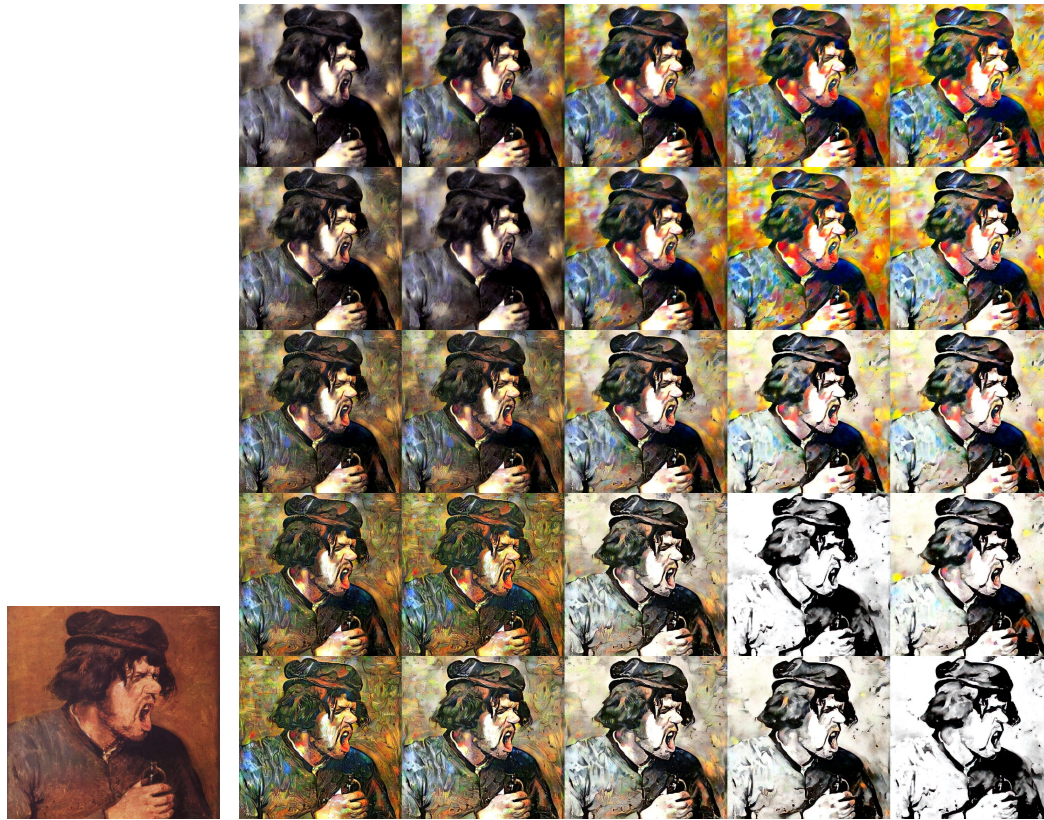

(a) Content image    (b) Pairwise interpolations between four freely chosen archetypal styles.

Figure 7: Free archetypal style manipulation of "The Bitter Drunk" by Adriaen Brouwer.

## 5 Discussion

In this work, we introduced archetypal style analysis as a means to identify styles in a collection of artworks without supervision, and to use them for the manipulation of artworks and photos. Whereas other techniques may be used for that purpose, archetypal analysis admits a dual interpretation which makes it particularly appropriate for the task: On the one hand, archetypes are represented as convex combinations of input image styles and are thus directly interpretable; on the other hand, an image style is approximated by a convex combination of archetypes allowing various kinds of visualizations. Besides, archetypal coefficients may be used to perform style manipulations.

One of the major challenge we want to address in future work is the exploitation of metadata available on the WikiArt repository (period, schools, art movement...) to link the learned styles to the descriptions employed outside the context of computer vision and graphics, which we believe will make them more useful beyond style manipulation.

### Acknowledgements

This work was supported by a grant from ANR (MACARON project ANR-14-CE23-0003-01), by the ERC grant number 714381 (SOLARIS project) and the ERC advanced grant Allegro.

## Footnotes

*Institute of Engineering Univ. Grenoble Alpes

[2] `https://github.com/sunshineatnoon/PytorchWCT`

[3] `https://github.com/rkjones4/GANGogh`

[4] `https://www.wikiart.org`

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
