[Supplementary Material · supplementary.pdf]

# Unsupervised Learning of Artistic Styles with Archetypal Style Analysis

# Supplementary Material

**Daan Wynen, Cordelia Schmid, Julien Mairal**
Univ. Grenoble Alpes, Inria, CNRS, Grenoble INP,*LJK, 38000 Grenoble, France
`firstname.lastname@inria.fr`

Here, we presents a set of additional results, which were not included in the paper for space limitation reasons, as well as experimental material such as the full set of archetypes learned by our approach.

## Contents

## 1   Influence of parameters $\gamma$, $\delta$ and comparison with [1].

In this section, we provide additional comparisons between our variant of [1] and the original one. All cases seem to confirm that (i) the heuristic $\gamma = \delta$ is reasonably good in terms of quality of the results, and (ii) our variant is much more accurate than [1] in terms of content preservation as soon as the amount of stylization is less than 100%. Figure 1 visualizes why the method of [1] loses so much detail regardless of the strength of stylization. Figures 2, 3, and 4 show more detailed comparisons of the two variants for different artworks and target styles.

Figure 1: Left: In [1] the original content image is used only once. While style information is injected before every decoding step, information lost during the encoding, coloring and decoding steps can never be recovered. Right: Using $I^c$ at every layer allows the decoders to produce more faithful reconstructions if desired.

delta_1.0_gamma_0.0.jpg    delta_1.0_gamma_0.25.jpg    delta_1.0_gamma_0.5.jpg    delta_1.0_gamma_0.75.jpg    delta_1.0_gamma_1.0.jpg

delta_0.75_gamma_0.0.jpg    delta_0.75_gamma_0.25.jpg    delta_0.75_gamma_0.5.jpg    delta_0.75_gamma_0.75.jpg    delta_0.75_gamma_1.0.jpg

delta_0.5_gamma_0.0.jpg    delta_0.5_gamma_0.25.jpg    delta_0.5_gamma_0.5.jpg    delta_0.5_gamma_0.75.jpg    delta_0.5_gamma_1.0.jpg

delta_0.25_gamma_0.0.jpg    delta_0.25_gamma_0.25.jpg    delta_0.25_gamma_0.5.jpg    delta_0.25_gamma_0.75.jpg    delta_0.25_gamma_1.0.jpg

delta_0.0_gamma_0.0.jpg    delta_0.0_gamma_0.25.jpg    delta_0.0_gamma_0.5.jpg    delta_0.0_gamma_0.75.jpg    delta_0.0_gamma_1.0.jpg

(a) Images produced by our approach when varying $\delta$ and $\gamma$.

(b) Images produced by our approach when $\gamma = \delta$, jointly increasing these parameters from 0 (left) to 1 (right).

baseline_0.0.jpg    baseline_0.25.jpg    baseline_0.5.jpg    baseline_0.75.jpg    baseline_1.0.jpg

(c) Images produced by the original approach of [1] when changing their stylization parameter.

Figure 2: Comparison of stylization control between our approach and [1].

delta_1.0_gamma_0.0.jpg delta_1.0_gamma_0.25.jpg delta_1.0_gamma_0.5.jpg delta_1.0_gamma_0.75.jpg delta_1.0_gamma_1.0.jpg

delta_0.75_gamma_0.0.jpg delta_0.75_gamma_0.25.jpg delta_0.75_gamma_0.5.jpg delta_0.75_gamma_0.75.jpg delta_0.75_gamma_1.0.jpg

delta_0.5_gamma_0.0.jpg delta_0.5_gamma_0.25.jpg delta_0.5_gamma_0.5.jpg delta_0.5_gamma_0.75.jpg delta_0.5_gamma_1.0.jpg

delta_0.25_gamma_0.0.jpg delta_0.25_gamma_0.25.jpg delta_0.25_gamma_0.5.jpg delta_0.25_gamma_0.75.jpg delta_0.25_gamma_1.0.jpg

delta_0.0_gamma_0.0.jpg delta_0.0_gamma_0.25.jpg delta_0.0_gamma_0.5.jpg delta_0.0_gamma_0.75.jpg delta_0.0_gamma_1.0.jpg

(a) Images produced by our approach when varying $\delta$ and $\gamma$.

(b) Images produced by our approach when $\gamma = \delta$, ranging from 0 (left) to 1 (right).

baseline_0.0.jpg baseline_0.25.jpg baseline_0.5.jpg baseline_0.75.jpg baseline_1.0.jpg

(c) Images produced by the original approach of [1].

Figure 3: Comparison of stylization control between our approach and [1].

(a) Images produced by our approach when varying $\delta$ and $\gamma$.

(b) Images produced by our approach when $\gamma = \delta$, jointly increasing these parameters from 0 (left) to 1 (right).

(c) Images produced by the original approach of [1] when changing their stylization parameter.

Figure 4: Comparison of stylization control between our approach and [1].

# 2 Examples of Image Decompositions

We show in this section a few additional image decompositions, involving trivial ones, meaningful ones, and failure cases.

(a)

(b) Image decomposition.

Figure 5: Image decompositions from the GanGogh collection.

(a)

(b) Image decomposition.

Figure 6: Image decompositions from the GanGogh collection.

(a)

(b) Image decomposition.

Figure 7: Image decompositions from the GanGogh collection.

(a)

(b) Image decomposition.

Figure 8: Image decompositions from the GanGogh collection.

(a) Four archetypal decompositions.

Arch. 248 = 100.00

Arch. 249 = 50.49 + 45.47 + 4.04

Arch. 250 = 100.00

Arch. 251 = 100.00

(b) Image decomposition.

Image 4400 ≈ 52.67 + 15.55 + 12.54

52.67 + : 18.17 · 9.48 · 7.76

15.55 + : 50.23 · 15.60 · 11.55

12.54 : 22.66 · 15.25 · 14.21

Figure 9: Failure cases of two archetypal decompositions (a) and image decomposition (b). (a): the second archetype seems to code only for "circle on rough canvas". While this is definitely the defining characteristic of the contributing images, it is not helpful for stylization. The other rows are examples of degenerate archetypes, *i.e.* archetypes with a single contribution. (b) A non-sparse image decomposition, hence difficult to interpret. The strongest three components seem to represent the absence of texture, but it is not clear what their contribution is to the image style.

# 3 Additional Examples of Style Manipulation

In this section, we present additional examples of style enhancement and interplation, as well as examples of stylization of natural photographs.

(a) "Woman with Book" by Pablo Picasso. From the GanGogh collection.

Figure 10: We demonstrate the enhancement of the two most prominent archetypal styles for different artworks. The middle panel shows a near-perfect reconstruction of the original content image in every case and uses parameters $\gamma, \delta = 0.5$. Then, we increase the relative weight of the strongest component towards the left, and of the second component towards the right. Simultaneously, we increase $\gamma$ and $\delta$ from $0.5$ in the middle panel to $0.95$ on the outside.

Figure 11: "Maria and Baby" by Robert Henri. Free archetypal combination.

Figure 12: Tübingen image

Figure 13: Golden Gate Bridge

Figure 14: Additional examples of style enhancements of van Gogh's works.

Figure 15: Additional examples of style enhancements of van Gogh's works.

# 4 Full Set of van Gogh's Archetypes

In this section, we present the $k = 32$ archetypes learned on the collection of Van Gogh's paintings; the archetypes seem to cover van Gogh's artistic development relatively accurately. The full set of archetypes is shown in Figures 16 and 17.

Arch. 0 = 47.76 + 28.54 + 23.71

Arch. 1 = 49.06 + 38.63 + 12.31

Arch. 2 = 61.69 + 16.99 + 11.09

Arch. 3 = 84.11 + 15.89

Arch. 4 = 39.23 + 24.11 + 23.51

(a) Archetypes 0 to 4

Arch. 5 = 100.00

Arch. 6 = 69.41 + 18.07 + 12.31

Arch. 7 = 77.23 + 14.42 + 8.35

Arch. 8 = 66.32 + 33.68

Arch. 9 = 40.14 + 17.28 + 13.16

(b) Archetypes 5 to 9

Arch. 10 = 62.07 + 18.13 + 11.55

Arch. 11 = 83.41 + 16.59

Arch. 12 = 63.32 + 31.16 + 4.54

Arch. 13 = 59.67 + 40.33

Arch. 14 = 64.85 + 20.57 + 4.92

(c) Archetypes 10 to 14

Arch. 15 = 100.00

Arch. 16 = 51.01 + 34.89 + 14.09

Arch. 17 = 47.34 + 26.79 + 15.89

Arch. 18 = 91.88 + 8.12

Arch. 19 = 39.00 + 19.24 + 15.21

(d) Archetypes 15 to 19

Figure 16: Archetypes 0 to 19

Arch. 20 = 100.00

Arch. 21 = 100.00

Arch. 22 = 87.33 + 12.67

Arch. 23 = 54.68 + 31.23 + 14.09

Arch. 24 = 44.51 + 27.23 + 9.85

(a) Archetypes 20 to 24

Arch. 25 = 52.43 + 23.44 + 20.20

Arch. 26 = 41.48 + 27.21 + 13.55

Arch. 27 = 37.62 + 26.50 + 14.07

Arch. 28 = 34.53 + 24.21 + 17.27

Arch. 29 = 35.06 + 24.83 + 18.87

(b) Archetypes 25 to 29

Arch. 30 = 73.03 + 23.56 + 3.41

Arch. 31 = 100.00

(c) Archetypes 30 to 31

Figure 17: Archetypes 20 to 31

# 5 Full Set of GanGogh Archetypes

In this section, we provide the full set of 256 archetypes learned on the Gangogh collection. Visualization is performed in the same way as in the main paper.

Arch. 0 = 14.51 + 11.66 + 11.41

Arch. 1 = 27.55 + 26.81 + 22.24

Arch. 2 = 19.18 + 12.63 + 11.76

Arch. 3 = 57.88 + 21.16 + 9.62

Arch. 4 = 32.60 + 13.00 + 10.12

(a) Archetypes 0 to 4

Arch. 5 = 25.97 + 15.30 + 12.39

Arch. 6 = 23.41 + 14.40 + 8.89

Arch. 7 = 77.69 + 13.56 + 8.75

Arch. 8 = 31.91 + 10.78 + 10.74

Arch. 9 = 100.00

(b) Archetypes 5 to 9

Arch. 10 = 32.64 + 20.09 + 14.79

Arch. 11 = 21.48 + 13.53 + 12.77

Arch. 12 = 20.67 + 10.64 + 8.48

Arch. 13 = 20.88 + 13.18 + 12.74

Arch. 14 = 26.86 + 19.07 + 15.07

(c) Archetypes 10 to 14

Arch. 15 = 25.97 + 19.71 + 16.69

Arch. 16 = 20.09 + 18.79 + 18.66

Arch. 17 = 26.48 + 14.20 + 13.70

Arch. 18 = 18.17 + 9.48 + 7.76

Arch. 19 = 37.80 + 31.76 + 16.68

(d) Archetypes 15 to 19

(a) Archetypes 20 to 24

Arch. 20 = 29.99 + 28.67 + 20.66
Arch. 21 = 19.78 + 18.63 + 17.54
Arch. 22 = 10.05 + 8.72 + 7.79
Arch. 23 = 15.90 + 14.78 + 14.70
Arch. 24 = 18.17 + 17.17 + 14.29

(b) Archetypes 25 to 29

Arch. 25 = 19.88 + 16.13 + 8.69
Arch. 26 = 51.54 + 48.46
Arch. 27 = 42.26 + 34.99 + 22.75
Arch. 28 = 33.07 + 26.76 + 12.46
Arch. 29 = 41.35 + 21.41 + 18.75

(c) Archetypes 30 to 34

Arch. 30 = 51.64 + 31.63 + 12.58
Arch. 31 = 31.97 + 16.40 + 14.65
Arch. 32 = 28.20 + 18.47 + 14.19
Arch. 33 = 26.66 + 15.41 + 11.61
Arch. 34 = 100.00

(d) Archetypes 35 to 39

Arch. 35 = 47.35 + 19.18 + 14.81
Arch. 36 = 21.35 + 13.41 + 11.54
Arch. 37 = 42.52 + 35.41 + 22.08
Arch. 38 = 15.70 + 13.75 + 13.54
Arch. 39 = 15.50 + 11.59 + 10.60

(a) Archetypes 40 to 44

Arch. 40 = 28.42 + 27.83 + 24.19
Arch. 41 = 91.15 + 3.82 + 2.72
Arch. 42 = 21.23 + 16.54 + 13.27
Arch. 43 = 44.77 + 28.78 + 26.45
Arch. 44 = 100.00

(b) Archetypes 45 to 49

Arch. 45 = 30.06 + 28.66 + 13.07
Arch. 46 = 23.91 + 14.30 + 10.74
Arch. 47 = 29.35 + 24.85 + 11.14
Arch. 48 = 100.00
Arch. 49 = 42.46 + 19.98 + 17.38

(c) Archetypes 50 to 54

Arch. 50 = 100.00
Arch. 51 = 49.60 + 21.06 + 16.42
Arch. 52 = 20.81 + 16.62 + 12.13
Arch. 53 = 29.67 + 27.75 + 18.77
Arch. 54 = 11.88 + 9.16 + 7.15

(d) Archetypes 55 to 59

Arch. 55 = 50.14 + 24.18 + 21.81
Arch. 56 = 100.00
Arch. 57 = 56.75 + 43.25
Arch. 58 = 100.00
Arch. 59 = 55.68 + 44.32

Arch. 60 = 100.00

Arch. 61 = 29.59 + 24.05 + 22.27

Arch. 62 = 37.70 + 31.15 + 16.09

Arch. 63 = 29.54 + 18.94 + 17.70

Arch. 64 = 38.88 + 22.52 + 21.76

(a) Archetypes 60 to 64

Arch. 65 = 22.03 + 13.47 + 13.06

Arch. 66 = 71.17 + 19.42 + 8.30

Arch. 67 = 22.72 + 13.91 + 13.38

Arch. 68 = 50.44 + 37.34 + 12.08

Arch. 69 = 100.00

(b) Archetypes 65 to 69

Arch. 70 = 100.00

Arch. 71 = 18.91 + 18.42 + 15.57

Arch. 72 = 55.30 + 34.03 + 6.41

Arch. 73 = 44.98 + 20.43 + 9.94

Arch. 74 = 43.82 + 27.04 + 18.94

(c) Archetypes 70 to 74

Arch. 75 = 44.29 + 28.42 + 24.03

Arch. 76 = 100.00

Arch. 77 = 33.33 + 26.42 + 17.99

Arch. 78 = 75.85 + 24.15

Arch. 79 = 69.01 + 30.99

(d) Archetypes 75 to 79

Arch. 80 = 55.47 + 44.53

Arch. 81 = 20.45 + 18.87 + 12.09

Arch. 82 = 55.71 + 24.96 + 15.71

Arch. 83 = 19.75 + 12.76 + 10.09

Arch. 84 = 100.00

(a) Archetypes 80 to 84

Arch. 85 = 52.65 + 44.54 + 2.81

Arch. 86 = 100.00

Arch. 87 = 100.00

Arch. 88 = 91.73 + 8.27

Arch. 89 = 51.40 + 24.63 + 18.60

(b) Archetypes 85 to 89

Arch. 90 = 22.94 + 17.06 + 14.31

Arch. 91 = 23.54 + 22.66 + 16.33

Arch. 92 = 100.00

Arch. 93 = 54.37 + 35.77 + 9.86

Arch. 94 = 18.26 + 13.41 + 10.90

(c) Archetypes 90 to 94

Arch. 95 = 100.00

Arch. 96 = 47.03 + 33.77 + 14.55

Arch. 97 = 47.39 + 36.16 + 8.12

Arch. 98 = 100.00

Arch. 99 = 22.66 + 15.25 + 14.21

(d) Archetypes 95 to 99

Arch. 100 = 70.56 + 16.72 + 4.88

Arch. 101 = 56.79 + 22.67 + 13.05

Arch. 102 = 57.20 + 20.78 + 10.06

Arch. 103 = 34.98 + 27.44 + 10.72

Arch. 104 = 14.99 + 9.26 + 8.70

(a) Archetypes 100 to 104

Arch. 105 = 53.80 + 46.20

Arch. 106 = 42.76 + 36.27 + 20.97

Arch. 107 = 13.33 + 12.30 + 10.80

Arch. 108 = 100.00

Arch. 109 = 20.62 + 16.11 + 11.65

(b) Archetypes 105 to 109

Arch. 110 = 35.73 + 26.41 + 10.97

Arch. 111 = 100.00

Arch. 112 = 64.95 + 15.30 + 10.42

Arch. 113 = 40.95 + 21.80 + 13.03

Arch. 114 = 8.75 + 7.96 + 6.82

(c) Archetypes 110 to 114

Arch. 115 = 92.91 + 7.09

Arch. 116 = 100.00

Arch. 117 = 21.12 + 19.24 + 10.09

Arch. 118 = 21.24 + 21.07 + 17.91

Arch. 119 = 33.68 + 18.25 + 11.07

(d) Archetypes 115 to 119

Arch. 120 = 35.17 + 20.51 + 19.54

Arch. 121 = 37.58 + 18.41 + 12.34

Arch. 122 = 19.60 + 18.21 + 18.17

Arch. 123 = 100.00

Arch. 124 = 17.25 + 17.07 + 13.70

(a) Archetypes 120 to 124

Arch. 125 = 100.00

Arch. 126 = 39.53 + 16.13 + 12.76

Arch. 127 = 14.29 + 11.93 + 11.91

Arch. 128 = 56.86 + 31.10 + 7.92

Arch. 129 = 73.67 + 26.33

(b) Archetypes 125 to 129

Arch. 130 = 21.94 + 20.85 + 12.99

Arch. 131 = 65.55 + 14.08 + 12.35

Arch. 132 = 37.29 + 32.70 + 19.36

Arch. 133 = 100.00

Arch. 134 = 30.58 + 28.36 + 18.29

(c) Archetypes 130 to 134

Arch. 135 = 54.15 + 22.02 + 8.13

Arch. 136 = 56.96 + 43.04

Arch. 137 = 27.08 + 22.82 + 13.20

Arch. 138 = 100.00

Arch. 139 = 99.93

(d) Archetypes 135 to 139

Arch. 140 = 100.00

Arch. 141 = 20.77 + 19.83 + 11.72

Arch. 142 = 25.54 + 25.07 + 16.79

Arch. 143 = 53.38 + 46.62

Arch. 144 = 32.31 + 19.08 + 14.92

(a) Archetypes 140 to 144

Arch. 145 = 86.05 + 13.95

Arch. 146 = 41.63 + 24.62 + 9.01

Arch. 147 = 44.81 + 32.17 + 22.24

Arch. 148 = 36.82 + 13.45 + 9.99

Arch. 149 = 74.05 + 25.95

(b) Archetypes 145 to 149

Arch. 150 = 9.55 + 8.31 + 7.31

Arch. 151 = 100.00

Arch. 152 = 31.92 + 31.12 + 28.24

Arch. 153 = 100.00

Arch. 154 = 100.00

(c) Archetypes 150 to 154

Arch. 155 = 12.96 + 10.76 + 9.96

Arch. 156 = 47.09 + 16.35 + 11.84

Arch. 157 = 37.15 + 32.12 + 21.69

Arch. 158 = 27.33 + 15.59 + 14.84

Arch. 159 = 100.00

(d) Archetypes 155 to 159

Arch. 160 = 39.27 + 21.12 + 12.72

Arch. 161 = 59.34 + 14.58 + 12.17

Arch. 162 = 100.00

Arch. 163 = 82.80 + 15.87 + 1.33

Arch. 164 = 37.41 + 12.15 + 11.28

(a) Archetypes 160 to 164

Arch. 165 = 20.81 + 19.29 + 19.12

Arch. 166 = 21.38 + 14.35 + 13.46

Arch. 167 = 100.00

Arch. 168 = 100.00

Arch. 169 = 100.00

(b) Archetypes 165 to 169

Arch. 170 = 76.62 + 14.63 + 4.43

Arch. 171 = 51.55 + 48.45

Arch. 172 = 74.30 + 25.70

Arch. 173 = 100.00

Arch. 174 = 33.86 + 24.28 + 15.33

(c) Archetypes 170 to 174

Arch. 175 = 100.00

Arch. 176 = 100.00

Arch. 177 = 22.86 + 17.41 + 11.01

Arch. 178 = 42.14 + 31.18 + 22.02

Arch. 179 = 28.67 + 22.59 + 19.94

(d) Archetypes 175 to 179

Arch. 180 = 24.90 + 18.46 + 17.41

Arch. 181 = 15.01 + 9.75 + 8.38

Arch. 182 = 52.27 + 37.27 + 10.47

Arch. 183 = 100.00

Arch. 184 = 50.98 + 39.58 + 9.44

(a) Archetypes 180 to 184

Arch. 185 = 41.56 + 22.17 + 17.36

Arch. 186 = 38.56 + 14.35 + 11.87

Arch. 187 = 100.00

Arch. 188 = 40.55 + 23.13 + 21.04

Arch. 189 = 35.91 + 24.30 + 18.44

(b) Archetypes 185 to 189

Arch. 190 = 37.22 + 33.31 + 29.47

Arch. 191 = 82.46 + 15.21 + 2.32

Arch. 192 = 100.00

Arch. 193 = 100.00

Arch. 194 = 100.00

(c) Archetypes 190 to 194

Arch. 195 = 33.01 + 31.93 + 20.51

Arch. 196 = 100.00

Arch. 197 = 49.75 + 35.69 + 7.20

Arch. 198 = 21.90 + 17.12 + 16.57

Arch. 199 = 18.32 + 17.61 + 13.34

(d) Archetypes 195 to 199

Arch. 200 =      72.17      + 15.48      + 12.35

Arch. 201 =      100.00

Arch. 202 =      71.18      + 16.92      + 11.46

Arch. 203 =      37.25      + 29.50      + 26.16

Arch. 204 =      45.90      + 22.13      +  8.04

(a) Archetypes 200 to 204

Arch. 205 =      12.93      + 11.38      + 10.89

Arch. 206 =      64.03      + 14.77      + 13.68

Arch. 207 =      100.00

Arch. 208 =      44.85      + 33.11      + 16.94

Arch. 209 =      46.19      + 20.85      + 20.69

(b) Archetypes 205 to 209

Arch. 210 =      45.17      + 38.50      + 15.03

Arch. 211 =      100.00

Arch. 212 =      100.00

Arch. 213 =      35.33      + 24.07      +  9.48

Arch. 214 =      100.00

(c) Archetypes 210 to 214

Arch. 215 =      100.00

Arch. 216 =      21.84      + 19.11      + 15.41

Arch. 217 =      32.44      + 25.11      + 15.08

Arch. 218 =      54.85      + 45.15

Arch. 219 =      74.54      + 25.46

(d) Archetypes 215 to 219

Arch. 220 = 28.43 + 17.20 + 12.58

Arch. 221 = 32.55 + 25.21 + 17.61

Arch. 222 = 100.00

Arch. 223 = 29.66 + 14.17 + 11.74

Arch. 224 = 52.63 + 25.34 + 8.16

(a) Archetypes 220 to 224

Arch. 225 = 52.10 + 47.90

Arch. 226 = 74.62 + 25.38

Arch. 227 = 31.84 + 23.46 + 12.89

Arch. 228 = 100.00

Arch. 229 = 24.34 + 20.29 + 12.47

(b) Archetypes 225 to 229

Arch. 230 = 56.75 + 15.40 + 13.16

Arch. 231 = 35.87 + 25.07 + 19.77

Arch. 232 = 40.67 + 32.15 + 12.10

Arch. 233 = 75.52 + 24.48

Arch. 234 = 29.57 + 23.26 + 16.13

(c) Archetypes 230 to 234

Arch. 235 = 19.29 + 15.54 + 14.85

Arch. 236 = 38.80 + 36.20 + 17.51

Arch. 237 = 32.04 + 31.82 + 15.41

Arch. 238 = 59.04 + 40.96

Arch. 239 = 100.00

(d) Archetypes 235 to 239

Arch. 240 = 50.23 + 15.60 + 11.55

Arch. 241 = 63.82 + 18.56 + 10.85

Arch. 242 = 63.38 + 23.24 + 6.61

Arch. 243 = 34.84 + 32.28 + 15.16

Arch. 244 = 100.00

(a) Archetypes 240 to 244

Arch. 245 = 37.01 + 34.05 + 23.30

Arch. 246 = 39.14 + 20.22 + 12.80

Arch. 247 = 92.49 + 7.51

Arch. 248 = 100.00

Arch. 249 = 50.49 + 45.47 + 4.04

(b) Archetypes 245 to 249

Arch. 250 = 100.00

Arch. 251 = 100.00

Arch. 252 = 91.02 + 8.98

Arch. 253 = 83.92 + 16.08

Arch. 254 = 95.32 + 4.68

(c) Archetypes 250 to 254

Arch. 255 = 25.25 + 16.68 + 13.61

(d) Archetypes 255 to 255

## Footnotes

*Institute of Engineering Univ. Grenoble Alpes

# References

[1] Y. Li, C. Fang, J. Yang, Z. Wang, X. Lu, and M.-H. Yang. Universal style transfer via feature transforms. In *Adv. Neural Information Processing Systems (NIPS)*, 2017.