[Reviews · NeurIPS 2018]

Reviewer 1



This paper proposes a methods for unsupervised learning of artistic styles and stylistic transfer via “Archetypal” style analysis. Specifically, the authors apply a well-known dimensionality reduction technique known as "archetypal analysis" to deep image feature representations, to extract a dictionary of artistic style archetypes. They then decompose each image as a convex combination of relevant archetypes, which allows them to manipulate images by tuning coefficient of decomposition. This is an interesting, well-written paper on an interesting subject. The authors conduct extensive experiments, and observe certain (albeit qualitative - see below) improvements over existing style transfer approach. My main concern is the limited novelty of the proposed method, which is essentially an application of a well-known approach (archetype learning) to the stylistic modeling problem. Another concern, not limited to this paper only but applicable for this line of work in general, is lack of objective evaluation measures. Namely, the evidence presented in the experiments are all qualitative and thus not amenable to quantitative and objective assessment. For instance, it would be nice to see some experiments showing that the archetypal composition of images are similar if they belong to the same artistic style and/or “school”, e.g., post-renaissance, cubism, etc. At the very least, the authors could have conducted a user study evaluation, similar to one p[resented in Ref.[11]. Other questions/comments: - For the stylistic modeling/transfer problem, what’s so special about archetype modeling vs other dimensionality reduction techniques (PCA, Factor analysis, etc). For instance, FA is the tool of choice when doing personality trait analysis, which seems a related problem in some sense.

Reviewer 2



The paper presents a novel approach to alter the artistic style of images. This is achieved by combining an unsupervised style transfer method [11] with archetypal analysis [3] to learn style representations of collections of paintings (style images). Archetypes are computed for GanGogh and Vincent van Gogh paintings to learn style characteristics, which allows different stylization effects by changing the latent space of the archetypical representation. Due to the archetypical style representation, style changes remain interpretable. The style transfer is done in a hierarchical fashion similar to [11] by matching the first and second order statistics of the content and style feature maps (introduced as whitening and coloring transformations in [11]). The paper is clearly written and mostly well structured. - No conclusion is drawn in the end of the paper. Some conclusion / discussion should be added. - An image similar to Fig 1. in [11] could improve describing the workflow (including the differences of Eq. 3 and Eq. 4). Implementation details are omitted in the paper but since the implementation will be made publicly available, it should be easy to reproduce the results. The presented paper combines familiar techniques (i.e. [11] and [3]) in a novel way and presents adequate advances over existing methods. While [11] only use the content image as input to the lowest level (i.e. the stylized content images are propagated to higher levels), the proposed method uses the content image for each level, which better preserves content information (i.e. content information altered by stylization of lower levels are lost otherwise). In contrast to [7], the representation of the styles using archetypes is better interpretable, this allows for better artistic control. While the quality stylization papers is difficult to judge due to the artistic component, the idea of combining archetypes and style transfer is clearly motivated, leading to a representation that allows to alter image style an interpretable fashion. Given the recent advances in style transfer in academic and industrial research, the tackled problem can be seen as important. While the individual parts are not novel (i.e. [11] and [3]), their combination leads to stylization approach with interpretable degrees of freedom that is worth being published. Given that the proposed method allows to alter artistic style in an interpretable way, and the implementation will be made publicly available, other researchers will likely build on this. Some general comments and questions: - Lines 88-90 state that the dimensionality is reduces with SVD, keeping more than 99% of the variance. Reporting the percentage of kept variance assumes a Gaussian distribution. However, the style representations are most likely not Gaussian distributed. - Why are there style artifacts visible in Fig 5. lower left image for [11] for gamma = 0 (i.e. artifacts in the faces are clearly visible)? For gamma = 0, (see Equation 4), the method should in each level encode and decode the content image and pass it to the next level. The style should not effect the output image - Line 179 how is one texture per archetype synthesized with the approach of [11]?

Reviewer 3



Summary: The authors introduce an unsupervised approach to learn structure in the space of artistic styles, which allows to represent an arbitrary artistic image as a combination of “basis” or “archetypal” styles, as well as change the input image (resynthesize it) by modifying a contribution of each of the archetypal styles. Following Gatys et al., an artistic style is defined as a set of first and second order statistics (concatenated in a vector) in layers of a pre-trained deep network (authors use VGG-19). The authors compute such parametric styles representations for a collection of artistic images, apply SVD for dimensionality reduction (to make vectors 4096-dim), and use archetypal analysis on the resulting data set of 4096-dim style representations. Given a data set X, the archetypal analysis method allows to learn a dictionary Z of vectors, allowing to approximately represent each x = \sum{\alpha_i * z_i}, where \alpha_i sum up to one (elements of a simplex). Dictionary vectors can themselves be decomposed z = \sum{\beta_i * x_i} in terms of data vectors as well. This allows to compute corresponding mean features maps and covariance matrices of conv layers for vectors in Z (by linearly combining those of x_i), which makes Z a dictionary of styles. For synthesis the authors suggest an extension to the method of Li et al., consisting of whitening and color transforms in the feature space of the DNN (using the means and covariance matrices of styles images), and trained encoder / decoder pairs for fast mapping to and from features spaces. The proposed extension add a parameter to explicitly model a trade-off between style and content lacking in the original model. Overall, for any image the method allows to find its decomposition into archetypal styles, visualize (using the synthesis) each of them as textures, have input image stylized using archetypal styles, and modify the style of the input image by changing the decomposition coefficients and resynthesizing it. Confidence score: I’m fairly familiar with the literature and practical implementations of style transfer, and I think I understood the proposed method relatively well. Quality: The paper is good quality containing thorough description of the method and quite extensive experimental results. Clarity: The paper is clearly written and easy to follow. Originality: The paper addresses the problem of decomposing and visualizing artistic styles, combing ideas from style transfer literature (universal style transfer) and unsupervised learning (archetypal analysis). Such analysis of styles is novel as it gives a way to discover structure in the space of styles, while prior work mostly focused on synthesis and not on decompositions of styles. Significance: The problem of decomposition of styles, or finding factors of variation in styles, is difficult. There are methods allowing to transfer only certain aspects of the style image (e.g. color, fine- or coarse-grained details, …), but a principled description of what the elementary components of a style are is missing. This submission is a step in this direction. Though the archetypal styles are themselves quite complex and could be difficult to interpret (it is not that each archetypal texture is either a color blob or a repeated brush stroke, or something else basic enough to form a “linearly independent basis“ in the space of styles), but nevertheless they provide a new way to describe styles, and the proposed methods allows for fine-grained modifications of input images, which is interesting for applications. Therefore I believe this submission should be accepted for publication.